# Cytomorphometric and Clinical Changes in Gingival Tissue after Subgingival Tooth Preparation—A Pilot Study

**DOI:** 10.3390/healthcare11030414

**Published:** 2023-02-01

**Authors:** Marija Jovanovic, Nikola Zivkovic, Nikola Gligorijevic, Marko Igic, Milica Petrovic, Marija Bojovic, Rodoljub Jovanovic, Milena Kostic

**Affiliations:** 1Department of Prosthodontics, Faculty of Medicine, University of Niš, 18000 Niš, Serbia; 2Department of Pathology, Faculty of Medicine, University of Niš, 18000 Niš, Serbia; 3Department of Oral Medicine and Periodontology, Faculty of Medicine, University of Niš, 18000 Niš, Serbia; 4Faculty of Medicine, University of Niš, 18000 Niš, Serbia

**Keywords:** fixed prosthodontic restorations, gingival index, gingival bleeding index, cytomorphometry

## Abstract

Tooth preparation for a metal-ceramic crown with a subgingival finish line can lead to inflammatory changes in the gingival tissue, often accompanied by cell damage. This study aimed to evaluate the clinical signs of inflammation and the cytomorphological status of the gingival tissue before and after tooth preparation. The research included a homogeneous group of 19 patients with an indication for upper canine preparation. Before and after treatment, the gingival and the gingival bleeding indexes were determined, gingival swabs were taken, and direct smears prepared on slides for cytomorphometric analysis. The values of the measured gingival indexes were statistically significantly higher (*p* < 0.001) after tooth preparation. They decreased over time, which indicated the reversibility of the resulting changes. Cytological examination showed no statistically significant difference between the values of nuclear area, perimeter, Feret diameter, Feret angle, integrated optical density, MinFeret, and roundness, before and after the treatment. Significantly higher values of circularity, integrated optical density, MinFeret (*p* < 0.05), as well as roundness (*p* < 0.001) were found after 72 h, compared to those taken 15 min after tooth crown preparation. This study is a pioneering attempt to show gingival changes during fixed prosthodontic treatment and may shed new light on pathogenetic events in prosthodontic patients.

## 1. Introduction

Partial edentulousness is one of the most common healthcare problems occurring in 75% of the population in many regions of the world [1]. Tooth loss can lead to oral and systemic diseases, simultaneously reducing the patient’s quality of life. Dealing with it not only as a health, but also as a socioeconomic problem, presents a great professional challenge [2,3]. 

The production of a fixed prosthodontic restoration is supposed to improve the function and aesthetics of the orofacial system, providing the patient with comfort, and acting preventively against possible damage to oral tissues. Tooth preparation for fixed prosthetic restorations involves the removal of the hard tooth structure to provide space for cementing their retainers, all with the aim of complete occlusal rehabilitation of the patient [4]. 

The maintenance of periodontal health at the margins of crowns represents a complex challenge for prosthodontists and margin location of the restoration in relation to the alveolar bone height is one of the important factors which ensures the long-term health of the gingiva [5,6]. Crowns and bridges may cause interference with host defenses and create areas of microbial contamination leading to bacterial biofilm formation and subsequent damage to the periodontium [7].

Periodontal tissues must be healthy before the onset of prosthodontic treatment, and additional periodontal care is commonly indicated to ensure the best prosthodontic treatment outcome [2]. Tooth preparation for fixed prosthodontic restoration, especially finish line subgingival tooth preparation, may cause changes in the gingival tissue in the form of hyperemia and inflammation [3]. Inflammation affects the free gingiva, the interdental papilla, and the coronal part of the attached gingiva. It is characterized by changes in the tissue and blood vessels, followed by exudation and swelling of the affected part of the gingiva. In proportion to the increase in cell volume, changes at the level of cell organelles and nuclei are also possible [8]. Cellular infiltration is also present, which causes edema [9]. 

The clinical assessment of the condition of the gingiva and the entire periodontium is carried out with indexes (gingival index and bleeding index) which enables numerical expression of the occurred changes, objective assessment of the condition of the periodontium, and monitoring of achieved therapeutic results and their precise comparison in epidemiological and scientific research [10,11]. 

Oral exfoliative cytology, which studies features of exfoliated cells from the oral mucosa, is a simple, non-aggressive, relatively painless technique that patients accept well. Quantitative techniques based on parameters such as variations in the size of the nucleus, as well as the cytoplasm, and alterations in the nucleus–cytoplasm ratio can increase the diagnostic sensitivity of exfoliative cytology [12]. The monitoring of morphological changes in gingival cells in comparison to clinical parameters after tooth preparation is a new aspect of dealing with potential inflammatory changes after tooth preparation, intending to improve this type of prosthodontic therapy and minimize possible damage.

The aim of this pilot study was to evaluate gingival indexes and the cytomorphological appearance of the gingival tissue before and after tooth preparation with subgingival margin placement.

## 2. Materials and Methods

### 2.1. Ethics

The Ethics Committee of the Clinic of Dental Medicine, Faculty of Medicine, University of Niš, approved the study protocol (Number: 68/5-2019-3EO) written in agreement with the principles of the Declaration of Helsinki. All patients signed an Informed Consent statement.

### 2.2. Subjects

The study included 19 patients of both sexes, aged 35 to 65, without oral and systemic diseases or prosthodontic restorations. None of the patients had tooth decay.

Exclusion criteria were the following: acute infections, chronic diseases, smoking, alcoholism, antibiotic and corticosteroid use in the last 6 months, and patients undergoing chemotherapy and radiotherapy. 

Before prosthodontic treatment, basic periodontal therapy was performed in all patients. After periodontal treatment, all patients have stable periodontal disease with PDI (Periodontal Disease Index) = 5 and CPITN (Community Periodontal Index of Treatment Needs) = 3. The WHO periodontal probe was used when determining periodontal status. 

Using these principles, we achieved homogeneity of the group, where each patient had the indication for a maxillary canine PFM crown preparation.

### 2.3. Clinical Procedure

Each of the subjects had a maxillary canine prepared for a PFM crown 15 min in duration by the same therapist. The finish line of the preparation was localized 0.2 to 0.3 mm in the gingival sulcus (subgingival demarcation). Although subgingival location of the finish line gives the best esthetic results, it also causes the biggest gingival tissue destruction. In almost half of all PFM crown cases, a subgingival finish line is indicated due to better aesthetics, as was the case in our study [13].

The preparation was performed following all postulates established in the literature, with minimal damage to the gingival tissue. By using round and tapered diamond burs, a chamfer finish line was created. This type of finish line preparation is conservative and provides adequate bulk for the metal–ceramic crown. Preparing this type of finish line for a fixed prosthodontic restoration enables less removal of dental tissue, adequate marginal integrity, and health of periodontal tissues, with great longevity and aesthetics [14]. 

Even though temporary resin crowns are widely used when performing fixed prosthodontic restorations, in this pilot study we refrained from using them in order to see the true effect of tooth preparation on gingival health, without the interference of temporary crowns, temporary resin dental materials, and cements [15,16,17]. For the same reason, dental impressions and usual gingival retraction procedures were not conducted until completing the third observation period (72 h) [10,18].

The gingival index (GI) was used to determine the condition of the gingiva. The measurement was performed on the axial surfaces of the teeth (vestibular, oral, and proximal sides). The total gingival index is obtained by summing index values from all sides of the teeth and then dividing the obtained value by 4. Numeric values from 0 to 3 determine the clinical condition of the gingiva.

The gingival bleeding index (GBI) was determined by probing the gingival sulcus with a blunt periodontal probe. The intensity of the resulting bleeding was ranked based on the state of the gingiva after probing: 0—no bleeding, 1—bleeding 10–30 s after probing, 2—bleeding during gingival probing, and 3—spontaneous bleeding of the gingiva.

Experimental parameters were obtained after the observation periods of 15 min and 24 and 72 h after tooth preparation and thus forming three experimental groups. Samples taken 5 min before tooth preparation formed the control parameters.

It is documented that the 72-h period is enough to adequately evaluate the acute clinical gingival status after tooth preparation [19]. After the third observation period, impressions and both laboratory and clinical procedures following tooth preparation for PFM crown production were conducted.

### 2.4. Cytomorphometry

Samples were collected by taking a gingival swab on the vestibular side of the treated tooth using a sterile swab stick. This minimal invasive method does not affect gingival changes caused by the tooth preparation [12]. 

The material was applied to a non-greased slide, distributed evenly in a thin layer, dried at room temperature, then fixed in 96% ethyl alcohol for 10 min, and stained with hematoxylin–eosin. After staining, the samples were covered with coverslips and glued with DPX glue.

Image J software version 1.52a, accessed on 23 April 2018 (http://imagej.nih.gov/ij/download.html) was used for morphometric analysis, together with an optical microscope (Nikon, ECLIPSE 50i, Tokyo, Japan) with a magnification of ×400. With Image J, epithelial cell nuclei were selected using the computer mouse. The smears were used to evaluate changes in cells that characterize the inflammatory process using the following nuclear variables in the cytological material: nuclear size (area), perimeter (perim), integrated optical density (IntDen), minimal value of Feret’s diameter (MinFeret), nuclear roundness (round), circularity (circ), Feret’s diameter (Feret), Feret’s angle, and solidity. Purpose of these parameters were already defined as potential inflammation markers of squamous epithelial cells. The mentioned parameters have already been defined as potential markers of inflammatory changes on the squamous epithelium of the gingiva, with the clear purpose of linking the morphological and physiological changes on the tissue after uniform tooth preparation in a homogeneous group of patients. [20,21].

Gingival swabs were taken before and after the treatment at four time points (5 min before and 15 min, 24 h, and 72 h after tooth preparation), in concurrence with observation periods for determining clinical markers of inflammation. Samples taken 5 min before tooth preparation were used as control parameters.

### 2.5. Statistics

The obtained data were statistically processed in the SPSS 20.0 program. Continuous variables were represented by arithmetic means (X), standard deviations (SD), and medians (Md). The Shapiro–Wilk test was used to determine the normality of the distribution of continuous variables. As the distribution of the examined parameters deviated from the normal, the comparison of the values of continuous variables of two repeated measurements within the examined samples was performed using the Wilcoxon signed-rank test. The Friedman test was used to determine the statistically significant difference of continuous variables in multiple repeated measurements, and the size of the time effect was defined by the Kendall concordance coefficient. A value of *p* < 0.05 was considered statistically significant. The Student’s *t*-test was used to compare values of continuous variables for dependent samples in cases of normal distribution, i.e., the Mann–Whitney test when sample distribution deviated from normal.

## 3. Results

The study included 19 patients, eight of them (42.1%) were males and 11 (57.9%) were females. The mean age of the group was 52.79 ± 10.76 years (from 35 to 65 years).

### 3.1. Gingival Indexes

The values of both tested parameters are highest 15 min after tooth preparation and decrease in the second and third experimental periods (24 h and 72 h, respectively) (Table 1, Figure 1). The Wilcoxon signed-rank test determined that the values of these parameters were statistically significantly higher in all periods after, compared to the values before tooth preparation. Furthermore, parameter values 24 h and 72 h after tooth preparation were statistically significantly lower compared to parameter values 15 min after. The Friedman test revealed a statistically significant difference in the values of the examined parameters during the entire study, and the influence of time, based on the value of the Kendall concordance coefficient, was statistically very high.

### 3.2. Cytomorphometry

Cytological examination showed no statistically significant differences in area, perim, Feret, and Feret angle values between the control and experimental parameters, as well as between each experimental parameter (Table 2).

Furthermore, there was no statistically significant difference in IntDen, MinFeret, and round.

Experimental parameter values of circ and solidity were statistically significantly higher in parameters obtained after 72 h compared to the control parameters (*p* < 0.05). Furthermore, statistically significant higher values of circ, IntDen, MinFeret (*p* < 0.05), as well as round (*p* < 0.001), were found after 72 h compared to the sample taken 15 min after prosthodontic treatment. The solidity value was statistically significantly higher in the group in which samples were taken after 72 h compared to the group in which samples were taken after 24 h (*p* < 0.05).

The morphometric analysis of the cytological material is shown in Figure 2.

## 4. Discussion

In general, researchers use pilot studies to evaluate the adequacy of their planned methods and procedures [22].

Periodontal diseases have a chronic inflammatory character and are the result of a response to present pathogenic microorganisms. Tooth preparation for fixed restorations may lead to mechanical damage to the gingiva, which is accompanied by exudation and swelling that provide a suitable ground for the penetration of microorganisms. Negligible inflammation of the gingiva, if not diagnosed on time and properly treated, may result in the destruction of deeper periodontal tissues with permanent consequences for oral health, accompanied by tooth loss. Their clinical course largely depends on the immune response of the host [19,23]. 

The study was based on the assumption that tooth preparation for a fixed restoration causes damage to the gingiva, which could, consecutively with further inadequate prosthetic treatment, lead to deeper damage of other periodontal tissues. The study was carried out in the form of a pilot study, with all its shortcomings, in order to obtain relevant results and conclusions that would properly direct the future clinical prospective study with the aim of using multidisciplinary methods to point out the challenges of this common clinical procedure and propose adequate preventive measures.

To reach objective conclusions, it was necessary to select a homogeneous group with healthy periodontium [9]. Therefore, the pilot study included 19 subjects with a similar clinical picture and an indication for a metal-ceramic crown on the upper canine. On the other hand, the age of the subjects empirically indicated the existence of certain changes in the periodontal tissues, mainly on the gingiva, and thus clinical observation of the subjects was carried out, followed by periodontal therapy [5]. Before tooth preparation, stable periodontal disease was confirmed with values of three for CPITN and five for PDI, which enabled the same starting position for all subjects included in the study [6,10].

The preparation of the maxillary canine in all subjects was performed by the same therapist with an identical set of dental burs lasting 15 min, as well as the selection of the chamfer finish line, which aimed to standardize the methodological procedures in the study. In this study group, subgingival finish line was indicated, with maximal preservation of gingival sulcus tissue integrity and attached epithelium. The subgingival position was chosen on the assumption that it causes the greatest damage to the gingiva, regardless of maximum compliance with the postulates of tooth preparation for fixed restorations described in the literature [24].

No gingival retraction method was performed after teeth preparation, as it has been proven to cause changes in the gingival tissue [25]. For the same reasons, an impression was not taken, nor were temporary resin crowns made, in order to completely avoid the unwanted influence of various procedures and the application of dental materials on the gingival tissue, and in order to obtain results that fully indicate the inflammatory effect of tooth preparation on the gingival tissue [15,16,18].

After tooth preparation, gingival indexes were determined as clinical indicators of changes in gingival tissue, in observation periods of 15 min, 24, and 72 h. After the last observation period, the standard prosthetic for the production of a PFM crown was continued, in order to restore function, comfort, and aesthetics to the patient as soon as possible. 

Prior to the onset of prosthodontic treatment, the values of the measured indexes amounted to 0, which indicated the absence of inflammatory changes in the gingiva [10]. The values of both gingival indexes (GI and GBI) were highest 15 min after tooth preparation (moderate inflammation), and statistically significantly decreased 24 h and 72 h after the treatment (mild inflammation). The obtained results suggest that the inflammatory changes caused by tooth preparation decrease over time, which indicates the reversibility of damage and a small possibility of permanent changes in the periodontal tissue. This study was conducted on humans, on a specially selected homogeneous group of subjects, which makes it unique in the literature so far. Considering all the limitations of the pilot study in terms of duration and sample size, the optimistic nature of the presented results could be supported by a longer duration study that would probably result in the complete healing of the gingiva, the lowest predicted index values for the assessment of the condition of the gingiva, and the absence of its recession.

The findings of other authors indicate that correct tooth preparation can protect the gingival tissue, regardless of the subgingival position of the finish line preparation [5,19,26].

Exfoliative oral cytology has shown the connection between the structure and function of the cells and provides an exact picture of the changes in the tissue that accompany clinical examinations before and after tooth preparation. This study is a pioneering attempt to show gingival changes during fixed prosthodontic treatment [21]. Considering such studies have not been done so far, a prospective study can be based on the results obtained in this study, which will compare several variables responsible for the health of the gingiva during the fixed restoration procedure. Very few studies have used exfoliative cytology to evaluate changes in gingival inflammation and periodontal disease.

The potential inflammatory effect of tooth preparation for fixed structure dentures was also analyzed through morphometric changes of epithelial cells. 

The study was based on the assumption that the changes caused by the described method of tooth preparation cause changes in the surface layers of the gingiva, and that epithelial cells can show deviations in shape and size.

Desquamated gingival cells belong to the superficial layer of the epithelium and they are irregular, polygonal, and transparent [27]. These gingival cells are easy to remove from the surface of the gingiva and can be microscopically examined to find certain structural changes that would indicate a potentially harmful effect of tooth preparation. 

A more extensive study and the inclusion of other types of tooth preparations could also analyze the pathohistological changes in the connective tissue.

The main advantages of exfoliative cytology are the following: the technique is rapid, relatively painless, easy to perform, and allows repeated sampling of biological material without destroying tissue integrity, which means it can be performed repeatedly in preventive screening programs as well as during routine dental examinations [28]. 

Cytology is essential in the assessment of oral inflammatory processes [27,29]. Yet, measuring cells and cell parts represents a great challenge for many researchers. The stratified squamous gingival epithelium is desquamated constantly and this desquamation depends on the mitotic activity of the basal layer of epithelial cells, enzymatic processes in epithelial cells, and mechanical irritations [30].

Cytomorphometry is a technique for analyzing cellular and nuclear characteristics, and it is applied both in the cytological and histological analysis of tissue samples. Moreover, it represents a sovereign method in the differential diagnosis of benign and malignant neoplasms of different localizations. There are only a few research papers about cytomorphometry and inflammation. In the available literature, there is no data on nucleus size and shape changes caused by mechanical trauma. Several authors described the increase in nuclear parameters as a consequence of inflammation, which correlates with the results of this study [31,32,33]. 

This study included cell nuclei only, yet an analogy can be made between the obtained results and the evidence of other authors on changes in cell volume associated with resulting inflammation. Namely, the cell nucleus is the most important regulatory part of the cell, it controls all cellular metabolic processes, and simultaneously monitors its morphological and structural changes. The results of this study suggest changes in the nucleus, with change in its the size. However, there are no results of other studies in this area. Nayar and Sundharam [31] analyzed normal buccal mucosa depending on age and gender. Their results suggest an increase in the size of the nucleus in women and in older age, which is most likely a consequence of a hormonal effect. On the other hand, there are studies on the cytomorphometry of the oral mucosa in various chronic-inflammatory processes. The results of those studies indicate a wide range of changes in terms of thickening, which is seen in the beginning, up to a high degree of atrophy in the advanced stages of the disease [29,32,33].

The aim of this study was to gain a better understanding of the gingival status before and after tooth preparation. It is known that the cell nuclei of the stratified squamous gingival epithelium become larger during inflammation of the gingiva [21], which is in accordance with the results of this study. The nuclear size was normal in the control group and in the experimental groups. This study is a pioneering attempt to show gingival changes during fixed prosthodontic treatment and may shed new light on pathogenetic events in prosthodontic patients. The association between morphological and physiological observations enables a better understanding of the inflammatory reactions in the superficial gingival epithelium.

These findings may shed new light on pathogenetic events in prosthodontic patients.

A study with longer observation periods of several months to a year would provide results which would describe the influence of tooth preparation on periodontal tissue. It is a minimally invasive method without possible negative effects on the structural stability of the restoration. Of course, epithelial healing in peri-implant tissues can be monitored using the same method [34].

Considering that this type of study on gingival tissue cells has not been published so far, it is conducted as a pilot study with the idea of comparing the finish line position of the prepared teeth in subsequent studies, all with the aim of obtaining more relevant results. The resulting changes are an unequivocal sign of the impact of the applied procedure on the gingival tissue. However, considering present methodological limitations and the lack of narrowly specialized literature data, it is necessary to conduct further investigations of morphometric analyses in dentistry.

## 5. Conclusions

The values of the gingival and the gingival bleeding indexes indicated a moderate inflammation of the gingiva 15 min after tooth preparation, which gradually, but statistically significantly, decreased in the second and third experimental periods (24 h and 72 h, respectively). Cytological examination showed no statistically significant differences among the values of area, perim, Feret, and Feret angle between the experimental and the control parameters, as well as in the same examined values. Statistically significantly higher values of circ, IntDen, MinFeret (*p* < 0.05) as well as round (*p* < 0.001) were noted after 72 h, compared to the group in which samples were taken 15 min after tooth preparation.

This pilot study provides the basis for future clinical prospective analysis. The comparison of clinical and cytomorphometric results in a larger group of patients and in a longer observation period will show the influence of different preparation techniques in the fixed prosthodontic restoration production on gingival and periodontal health. We hope that the larger study would provide useful recommendations for dentists so they could achieve the maximum therapeutic effect and minimal periodontal tissue damage.

## Figures and Tables

**Figure 1 healthcare-11-00414-f001:**
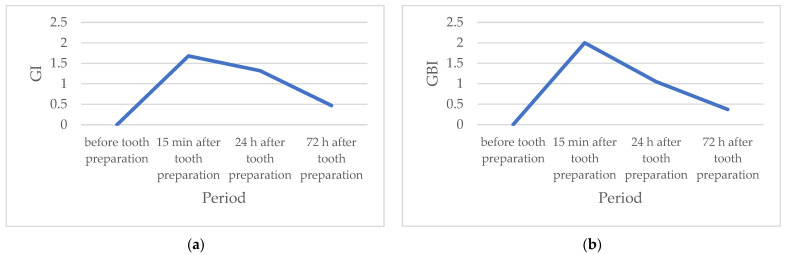
(**a**) Mean values of the gingival index (GI) during the study period; (**b**) Mean values of the gingival bleeding index (GBI) during the study period.

**Figure 2 healthcare-11-00414-f002:**
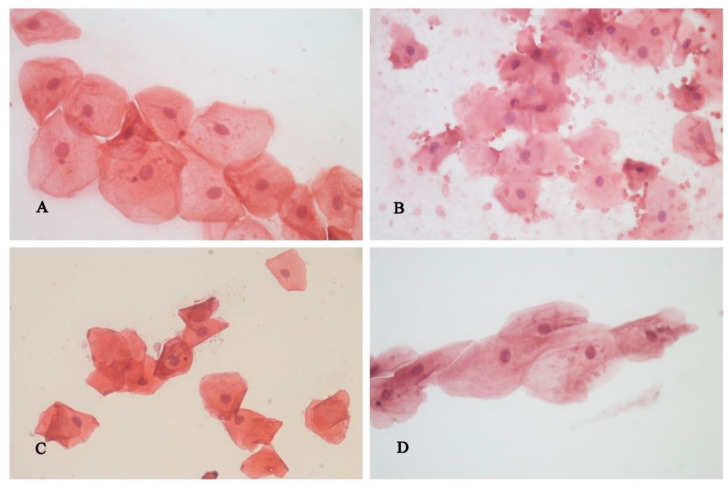
Superficial cells of the squamous gingival epithelium, cytological smear, H&E: (**A**)—before tooth preparation; (**B**)—15 min after tooth preparation, (**C**)—24 h after tooth preparation, (**D**)—72 h after tooth preparation.

**Table 1 healthcare-11-00414-t001:** GI and GBI values during the study period.

Period	GI			GBI		
a	0.00 ±	0.00	(0.00)	0.00 ±	0.00	(0.00)
b	1.68 ±	0.48	(2.00) ^a^ **	2.00 ±	0.47	(2.00) ^a^ ***
c	1.32 ±	0.48	(1.00) ^a^ *** ^b^ *	1.05 ±	0.52	(1.00) ^a,b^ ***
d	0.47 ±	0.51	(0.00) ^a^ ** ^b,c^ ***	0.37 ±	0.50	(0.00) ^a^ ** ^b,c^ ***
*p* (Friedman Test)	<0.001			<0.001		
Kendall’s Coefficient of Concordance	0.86			0.90		

Parameters are given as means ± standard deviation (median). a—5 min before tooth preparation, b—15 min after tooth preparation, c—24 h after tooth preparation, d—72 h after tooth preparation (Wilcoxon signed-rank test). *—*p* < 0.05, **—*p* < 0.01, ***—*p* < 0.001.

**Table 2 healthcare-11-00414-t002:** Cytological values during the study.

					* **p** *	
**Area**	**X**	**±SD**	**(Me)**	**vs. a**	**vs. b**	**vs. c**
5 min before	74.44	±18.46	(72.38)			
15 min after	68.75	±15.58	(68.46)	0.4846		
24 h after	71.13	±19.09	(66.64)	0.6048	0.9736	
72 h after	80.54	±23.99	(69.61)	0.5346	0.2465	0.2582
**Perim**	**X**	**±SD**	**(Me)**	**vs. a**	**vs. b**	**vs. c**
5 min before	32.22	±3.71	(31.32)			
15 min after	30.95	±3.36	(30.96)	0.5434		
24 h after	31.35	±3.71	(30.70)	0.5016	0.9868	
72 h after	32.95	±4.49	(30.80)	0.6538	0.4078	0.3461
**Circ**	**X**	**±SD**	**(Me)**	**vs. a**	**vs. b**	**vs. c**
5 min before	0.88	±0.04	(0.89)			
15 min after	0.89	±0.03	(0.89)	0.7710		
24 h after	0.89	±0.03	(0.89)	0.6401	0.7885	
72 h after	0.91	±0.03 ^ab^ *	(0.91)	0.0498	0.0279	0.1088
**Feret**	**X**	**±SD**	**(Me)**	**vs. a**	**vs. b**	**vs. c**
5 min before	12.15	±1.31	(11.87)			
15 min after	11.78	±1.30	(11.75)	0.3965		
24 h after	11.77	±1.38	(11.36)	0.2695	0.5962	
72 h after	12.11	±1.49	(11.66)	0.9416	0.4853	0.4287
**IntDen**	**X**	**±SD**	**(Me)**	**vs. a**	**vs. b**	**vs. c**
5 min before	21.87	±6.38	(22.10)			
15 min after	18.77	±3.88	(18.13)	0.0802		
24 h after	19.80	±5.56	(18.63)	0.3515	0.8167	
72 h after	21.99	±5.45 ^b^ *	(21.58)	0.9563	0.0497	0.1416
**FeretAngle**	**X**	**±SD**	**(Me)**	**vs. a**	**vs. b**	**vs. c**
5 min before	108.31	±24.61	(114.47)			
15 min after	102.19	±24.10	(100.69)	0.4500		
24 h after	105.02	±24.13	(107.24)	0.6971	0.7318	
72 h after	92.88	±23.06	(84.98)	0.0693	0.2538	0.1561
**MinFeret**	**X**	**±SD**	**(Me)**	**vs. a**	**vs. b**	**vs. c**
5 min before	8.31	±1.33	(8.11)			
15 min after	7.83	±1.07	(7.95)	0.2270		
24 h after	8.27	±1.17	(8.03)	0.9190	0.2499	
72 h after	8.92	±1.51 ^b^ *	(8.37)	0.2142	0.0370	0.2278
**Round**	**X**	**±SD**	**(Me)**	**vs. a**	**vs. b**	**vs. c**
5 min before	0.72	±0.10	(0.76)			
15 min after	0.70	±0.06	(0.71)	0.2127		
24 h after	0.73	±0.08	(0.74)	0.9039	0.1304	
72 h after	0.77	±0.04 ^b^ ***	(0.77)	0.3515	0.0005	0.1255
**Solidity**	**X**	**±SD**	**(Me)**	**vs. a**	**vs. b**	**vs. c**
5 min before	0.95	±0.01	(0.96)			
15 min after	0.96	±0.01	(0.96)	0.3399		
24 h after	0.95	±0.01	(0.95)	0.8035	0.2406	
72 h after	0.96	±0.01 ^ac^ *	(0.96)	0.0479	0.3696	0.0246

Continuous variables are given as means ± standard deviation (SD, median). ^a^—vs. control (5 min before tooth preparation), ^b^—vs. parameter 15 min after tooth preparation, ^c^—vs. parameter 24 h after tooth preparation. *—*p* < 0.05, ***—*p* < 0.001 (Student’s *t*-test of dependent samples/Mann–Whitney test).

## Data Availability

The data presented in this study are available on request from the corresponding author.

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
