# Peer review of "Cytomorphometric and Clinical Changes in Gingival Tissue after Subgingival Tooth Preparation—A Pilot Study"

_healthcare, 2023, doi:10.3390/healthcare11030414_

Round 1
Reviewer 1 Report
Dear Authors, the article is well written but
- I think there are many errors in the methodology .
- After tooth preparation, why did you not use a resin temporary crown? Do you think that phase could change the result ( I think yes...and it's the clinical reality....)
- What type of margin preparation did you use?
- why did you do last control after 72 hours...?
Dear Authors , I am sorry but this article must be completely changed. Best regards.
Author Response
Dear Reviewer,
Thank you for your time and detailed analysis of our manuscript.
In accordance with your and the suggestions from remaining reviewers, we tried to improve the manuscript and answer your comments, with the aim of presenting the obtained results completely and concisely.
- According to your recommendations, the methodology was explained in more detail, which significantly changed the form of the manuscript. A pilot study involves a limited number of patients, but was crucial for planning a future clinical prospective study. The modified methodology in the manuscript is detailed from line 99 up to line 116, and 130-133. The cytomorphometric procedure is described from lines 135-137.
- Concerning the topic of the production of temporary resin crowns, we avoided them due to the possible influence on the obtained results. The same applies to the retraction method and impression taking in general. Namely, these certainly important and necessary clinical procedures in fixed prosthodontic restoration fabrication can lead to gingival tissue changes, which has been proven by in vitro and in vivo studies and confirmed by references.
- We used the chamfer finish line with subgingival position for PFM crown preparation. This finish line position was chosen because it is assumed to cause the greatest changes in the gingival tissue. Certainly, in the upcoming clinical prospective study, we will compare different finish lines with different positions in relation to the gingival margin. It is also very important to examine all stages of the production of a fixed prosthodontic restoration on the gingival tissue, which unambiguously includes the making of temporary crowns as well. In this way, a greater number of results will be obtained and relevant conclusions will be drawn by comparing clinical and exfoliative cytology results.
- We planned the study to last 72 hours since it was envisioned as a pilot study and as such, it has its limitations. On the other hand, it was necessary to finish the patient's fixed restoration as soon as possible andrestore their function, aesthetics and comfort. The obtained results from this study are the basis for planning a future research with longer observation periods, lasting for several months or even several years. The emphasis will be primarily on cytomorphometry, because the data obtained from this pilot study indicates that cytomorphometry can be used in a purposeful way in dentistry, and also, it has not been done in prosthodontics so far. We found that time period of 72 hours coincided with the onset of gingival tissue repair after tooth preparation.
The manuscript was significantly expanded, additional explanations of the methodology were given, in the section on materials and methods, and in the discussion, so the presented results are clearer.
The conclusions have been rewritten in accordance to the changes in text with the clear statement that additional research is needed for better understanding of gingival tissue changes and introducing cytomorphometry as a relevant research tool in prosthodontics.
If you have any more questions or suggestions, we would gladly take them into consideration.

Reviewer 2 Report
In connection with periodontal examination, has a regular periodontal probe or a WHO periodontal probe been used?
The bibliographic references number 1, 5, 7, 12, 19, 20, 21 are 20 years old or older, more updated articles should be presented.
Original article, well written, that with minimal suggested changes can be published successfully.

Author Response
Dear Reviewer,
Thank you for your time and detailed analysis of our manuscript.
In accordance with your and the suggestions from other reviewers, we tried to improve the manuscript and answer your comments, with the aim of presenting the obtained results completely and concisely.
- The WHO periodontal probe was used when determining periodontal status, which is emphasized in the text.
- References 1, 5, 7 and 12 have been replaced by more recent references. We were not able to replace references 20 and 21 with new ones, because we refer to them in the text. Namely, cytomorphometry is a new examination method in dentistry, and there are no similar literature data. We hope that in our research we will introduce cytomorphometry as easily performed method that gives a clear insight into how the periodontium is affected by everyday prostodontic procedures, by combining the structural and functional characteristics of tissues.
If you have any more questions or suggestions, we would gladly take them into consideration.

Reviewer 3 Report
nice paper, conclusions are redundant and must be shorter.
Author Response
Dear Reviewer,
We would like to thank you for your time and detailed analysis of our manuscript and for the positive evaluations. In accordance with your suggestions, we shortened the conclusion.

Reviewer 4 Report
This is an interesting article using smears from gingival swab samples to evaluate tissue response post tooth preparation
What is the basis of selecting each of these cytomorphometric parameters. Please mention.
Discussion can include other articles which look at epithelial healing in peri-implant tissues
As only the superficial epithelium is scraped, it may not indicate complete healing and events in connective tissue
Were repeated smears taken from same sites for the different time intervals. Does this affect the epithelial healing and compound the inflammation in already injured tissue. Please clarify.
Line 112 - tree - correct as three
Line 166- beetwen each experimental parameters - Correct statement
Is it feasible to do a longer assessment at 1 month, 3 months, 6 months and 12 months using this minimally invasive method. Kindly discuss.
Author Response
Dear Reviewer,
Thank you for your time and detailed analysis of our manuscript.
In accordance with your and the suggestions from remaining reviewers, we tried to improve the manuscript and answer your comments, with the aim of presenting the obtained results completely and concisely.
- Cytomorphometry aims to explain the changes occurring in the tissue by looking at the cell structure and in conjunction with the clinically obtained gingival indexes provide more detailed results on the health of the gingiva. Given that these studies are extremely rare in dentistry, especially in prosthodontics, we chose the parameters based on the few available references in which the relevant results were obtained (references 20 and 21).
- The discussion included epithelial healing in peri-implant tissues (reference 34 - Ramenzoni, LL.; Attin, T.; Schmidlin, PR. In Vitro Effect of Modified Polyetheretherketone (PEEK) Implant Abutments on Human Gingival Epithelial Keratinocytes Migration and Proliferation. Materials. 2019, 12, 1401.)
- The aim of the study was to examine changes in the epithelium as the surface layer of the gingiva. The study proved the reversible nature of gingival inflammation and rapid tissue healing. Deeper changes and prolonged inflammation would include the connective tissue. However, that was not investigated in our study.
- Line 112 - three - corrected
- Line 166 - between - corrected
- Cytomorphometry is a minimally invasive method, and study with longer observation periods of several months to a year are planed (line 341).
If you have any more questions or suggestions, we would gladly take them into consideration.
